# Gastrointestinal Microbiota and Their Manipulation for Improved Growth and Performance in Chickens

**DOI:** 10.3390/foods11101401

**Published:** 2022-05-12

**Authors:** Shahna Fathima, Revathi Shanmugasundaram, Daniel Adams, Ramesh K. Selvaraj

**Affiliations:** 1Department of Poultry Science, The University of Georgia, Athens, GA 30605, USA; shahna.fathima@uga.edu (S.F.); dr.daniel.adams.ii@gmail.com (D.A.); selvaraj@uga.edu (R.K.S.); 2Toxicology and Mycotoxin Research Unit, US National Poultry Research Center, Athens, GA 30605, USA

**Keywords:** probiotic, prebiotic, synbiotic, poultry, microbiota

## Abstract

The gut of warm-blooded animals is colonized by microbes possibly constituting at least 100 times more genetic material of microbial cells than that of the somatic cells of the host. These microbes have a profound effect on several physiological functions ranging from energy metabolism to the immune response of the host, particularly those associated with the gut immune system. The gut of a newly hatched chick is typically sterile but is rapidly colonized by microbes in the environment, undergoing cycles of development. Several factors such as diet, region of the gastrointestinal tract, housing, environment, and genetics can influence the microbial composition of an individual bird and can confer a distinctive microbiome signature to the individual bird. The microbial composition can be modified by the supplementation of probiotics, prebiotics, or synbiotics. Supplementing these additives can prevent dysbiosis caused by stress factors such as infection, heat stress, and toxins that cause dysbiosis. The mechanism of action and beneficial effects of probiotics vary depending on the strains used. However, it is difficult to establish a relationship between the gut microbiome and host health and productivity due to high variability between flocks due to environmental, nutritional, and host factors. This review compiles information on the gut microbiota, dysbiosis, and additives such as probiotics, postbiotics, prebiotics, and synbiotics, which are capable of modifying gut microbiota and elaborates on the interaction of these additives with chicken gut commensals, immune system, and their consequent effects on health and productivity. Factors to be considered and the unexplored potential of genetic engineering of poultry probiotics in addressing public health concerns and zoonosis associated with the poultry industry are discussed.

## 1. Introduction

Over the last three decades, the poultry industry has expanded tremendously with an annual growth rate of over 5% when compared to the bovine and swine industries, which grew at 1.5% and 3%, respectively [1]. Such intense growth demands effective disease prevention and control strategies. Economic loss because of loss in production associated with disease outbreaks accounts for 20% of the value of poultry production [2]. The regulation on the use of antibiotics necessitated the need to find potential alternatives to replace antibiotic growth promoters and reduce the prevalence of bacterial diseases in organic livestock production. Some of the alternatives which have gained research interest are probiotics, prebiotics, postbiotics, organic acids, and plant extracts [3]. Probiotics emerged as a potential alternative to antibiotics because probiotics are primarily of host origin and are considered Generally Recognized As Safe (GRAS) [3]. Probiotics are commensal bacteria and are typically isolated from the gastrointestinal tract of healthy chickens, screened for probiotic characteristics in vitro first, followed by in vivo characterization.

The microbes in the gastrointestinal tract play a crucial role in promoting the health and production performances of chickens. The diversity of gut microbiota varies by region, with cecum having the highest microbial diversity [4]. The gut microbiota functions to improve the utilization of nutrients, prevent pathogen colonization, improve growth performance and metabolize mycotoxins in the feed [5,6,7]. It is important to monitor the gut microbiota in chickens as some of the gut microbiota such as *Salmonella* and *Campylobacter* cause zoonotic diseases in humans [8]. The early colonization of the gut by microbes is crucial as it can affect the morphology and physiology of the intestine and susceptibility to infections. In ovo inoculation of probiotics and prebiotics favors the early colonization of the gut by ‘beneficial’ microbiota post-hatch [9,10,11].

The gut barrier is comprised of intestinal microbiota and their metabolites, mucins secreted by goblet cells, host-derived antimicrobial peptides such as defensins, and cathelicidins, IgA, intestinal epithelium, microfold cells (M cells), Paneth cells, tuft cells and lymphoid tissues in the sub-epithelium and lamina propria [12,13]. The gut barrier serves to contain the gut microbiota within the lumen while permitting the absorption of nutrients [12,14]. Intestinal health, tolerance to food and microbial antigens, and homeostasis are achieved through complex interactions between the multiple components in the gut [13]. This review will provide an in-depth analysis of the current knowledge on prebiotics, probiotics, postbiotics, and synbiotics supplementation in poultry. The effects of supplementing probiotics, postbiotics, prebiotics, and synbiotics and the mechanisms through which these additives exert beneficial effects to modulate the gut microbiota, and the interaction of the gut microbiota with the growth, performance, and immunity in chickens are included. This review also addresses the potential disadvantages of probiotic application in poultry. This review highlights some of the challenges in probiotic supplementation in poultry, identifies gaps in our knowledge, and identifies new avenues to develop the next generation of effective probiotic products.

## 2. Development of Gut Microbiota

Intestinal epithelial cells, microbial community, and the immune system are components of the gut ecosystem [14]. Prior to large-scale incubation of eggs in incubators, the eggs were in contact with the nest or hen during the incubation period and thereby ensuring the vertical transmission of the maternal microbiota to chicks [15]. However, in commercial hatchery settings, chicks are hatched in a clean environment with no contact with the hen. Hence, the gut microbiota of newly hatched chicks is entirely dependent on environmental sources which can lead to a decrease in microbial diversity [16] and an increase in foodborne pathogen colonization of the gut. Transferring the gut microbiota of healthy adult chickens to day-old chicks will be a future approach to controlling foodborne pathogens in poultry.

Microorganisms colonize the gut mucosal epithelium, gut lumen, and caeca of chickens [17]. The microbial composition and diversity are influenced by age of the bird, region of the gastrointestinal tract, genotype of the bird, housing conditions, and feed composition [18,19]. Conventional microbiological methods limited the research and identification of the individual components of the chicken gut microbiota. The development of 16S rRNA-based next-generation sequencing tools has made it possible to characterize the gut microbiota of chickens [20,21].

Chicks can acquire microbiota at the embryonic stage during egg formation in the oviduct and during transport through the reproductive tract [22]. Post-hatch microbial acquisition is dependent on various factors such as production practices, diet, and environment. With the modernization of chicken production in large-scale hatcheries, the natural vertical transfer of microbiota from the hen is considerably diminished. Nevertheless, the passage of eggs through the reproductive tract and cloaca will deposit microbiota on the eggshell in the form of a complex biofilm [23]. The obligates anaerobes in the eggshell biofilm can survive the incubation period and can shape the microbial population of the chick. Hatchery sanitation protocols such as washing, fumigation, and chemical disinfection of the egg, though, will reduce the vertical transfer of microbiota. In a study conducted by Olsen et al. [24], it was demonstrated that the bacterial load on eggs decreased from 1 × 10^4^ to 1 × 10^5^ CFU to less than 10 CFU per egg. Although disinfection practices are inevitable to ensure optimal hatchability in commercial hatcheries, disinfection impedes the vertical transfer of microbiota. Hence chicks acquire a considerable proportion of gut microbiota from the environment post-hatch. 

The first species to colonize the chicken gastrointestinal tract are coliforms and fecal *Streptococci*, which are abundant by day 3 post-hatch [17,25]. The small intestinal microbiota is established by approximately 2 weeks of age. By day 40, *Lactobacillus* dominates the small intestinal microflora. Cecal microbiota is established by 6–7 weeks and is dominated by facultative and obligate anaerobic microbes, consisting of *Clostridia*, enterobacteria, fecal *Streptococci*, *Pediococci*, and *Pseudomonas aeruginosa* [17]. Microbial composition and complexity increase in the distal digestive tract leading to the fluctuation in fecal microbial composition [21]. The microbial density and the predominant species in the different regions of the gut are summarized in Figure 1.

## 3. Microbiota in the Crop

The crop harbors a large bacterial community consisting of bacterial cells in the order of 1 × 10^8^ to 1 × 10^9^ CFU g^−1^. The crop is colonized predominantly by gram-positive bacteria such as *Lactobacillus* spp. [26]. Other bacterial species isolated from the crop include *Bifidobacterium*, *Klebsiella pneumoniae*, *K. ozaenae*, *Escherichia coli*, *E. fergusonii*, *Enterobacter aerogenes*, *Eubacterium* spp., *Pseudomonas aeruginosa*, *Micrococcus luteus*, *Staphylococcus lentus*, and *Sarcina* spp. [27,28]. Crop microbiota ferments dietary fiber to short-chain fatty acids (SCFA). Acetate is the major SCFA in the crop [29]. SCFA decreases the pH of the crop to inhibit the growth of those pathogens that colonize and proliferate at neutral or slightly alkaline pH [30].

## 4. Microbiota in the Proventriculus and Gizzard

The proventriculus and gizzard have acidic pH which is not ideal for microbial colonization [31]. Gastric acid can penetrate the cell membrane of microbes resulting in decreased intracellular pH and disruption of the transmembrane proton motive force [32]. Similarly, lactic acid and acetic acid prevent the colonization of pathogens that are sensitive to acidic pH [33]. *Lactobacilli* is the dominant species in proventriculus and gizzard. Lactose-negative enterobacteria, enterococci, and coliform bacteria are also prevalent in the proventriculus and gizzard [20,25,34]. The bacterial concentration in the gizzard is similar to that of the crop, however, the bacterial fermentation is hindered by the acidic pH resulting in decreased acetate and lactate concentrations in the gizzard [27].

## 5. Microbiota in the Small Intestine

The concentration of bacteria in the small intestine is approximately 1 × 10^5^ CFU g^−1^. The microbial density is low in the duodenum due to the low pH, presence of digestive enzymes, high concentrations of bile salts, high oxygen pressure, and pancreatic secretions [31,35]. The conditions are favorable for microbial growth in the distal small intestine. The small intestine is colonized by *Lactobacillus*, *Turicibacter*, *Enterococcus*, *Clostridia*, and *Streptococci* [25,27,35]. The predominant SCFA in the small intestine is acetate, although the concentration is considerably lower than that in the caeca [27].

## 6. Microbiota in the Caeca

Caeca has the highest microbial density of 1 × 10^11^ CFU g^−1^ and is comprised mainly of obligate anaerobes of the phyla Firmicutes and Bacteroidetes [36]. Phyla Actinobacteria and Proteobacteria contribute to 2–3% of the cecal microbiota [25]. The cecal microbiota is comprised of gram-positive cocci, *Bifidobacterium* spp., *Clostridium* spp., *E. coli*, *Lactobacillus* spp., *Streptococcus* spp., and *Bacteroides* spp. [37]. The colonic microbiota is almost identical to that of the cecal microbiota and might contain ileal microbiota [25]. The SCFA composition of caeca can vary significantly based on the diet [27]. A study conducted by Walugembe, et al. [38] reported that increasing the insoluble dietary fiber increased the relative concentration of butyric acid and decreased the relative concentration of acetic acid. A pea-containing diet can increase the SCFA concentrations, particularly the acetate and butyrate concentration in the ceca [39]. In a study conducted by Wielen et al. [40], a negative correlation was observed between the concentrations of undissociated forms of acetate, propionate, and butyrate, and Enterobacteriaceae in caeca.

## 7. Dysbiosis

Dysbacteriosis or dysbiosis is defined as an undesirable alteration of the microbiota resulting in an imbalance between protective and harmful bacteria. The host harbors opportunistic microbes which under certain circumstances will overgrow and become pathogenic to the host. It is important to maintain homeostasis in the gut by preventing the entry of pathogenic microbes while preserving the balance between the existing commensal gut microbiota. The first line of defense against the entry of microbes into the submucosa is the mucin, a glycoprotein polymer secreted by the goblet cells. The outer loose layer of mucin acts as a substrate for the attachment and colonization of the bacteria while the inner compact layer does not permit attachment and colonization of most bacteria [41,42,43]. Gut microbiota regulates the secretion of mucus [44,45] and hence, dysbiosis can induce disruption of the mucin layer [46]. The microbes are separated from the sterile submucosa by a layer of selectively permeable intestinal epithelial cells (IECs) and adjoining tight junctions. The IECs secrete antimicrobial peptides such as defensins, cathelicidins, and C- type lectins to protect the host from the pathogens in the gut. Loss of submucosal integrity or loss of tight junction proteins such as claudins, occludins, and zona occludens can disrupt the barrier leading to the entry of microorganisms from the lumen into the submucosa and ultimately into the systemic circulation [47]. IECs express pattern recognition receptors (PRRs) such as toll-like receptors (TLRs) and nod-like receptors (NLRs) which are regulated by compartmentalization and a higher activation threshold. These receptors are activated by the antigens that breach the intestinal epithelial barrier [48]. The microbial-associated molecular patterns such as lipopolysaccharide, flagellin, and peptidoglycan stimulate the immune system, induce pro-inflammatory signaling, and induce inflammation which will act to eliminate the pathogens [48,49]. To summarize, dysbiosis is prevented by limiting the contact of pathogens with the intestinal epithelium and rapid detection and elimination of pathogens that breach the epithelial barrier.

The gut microbiota lies at the barrier of the internal and external environment of the gut and plays an important role in preventing gut dysbiosis [50]. The gut microbiota ferments complex carbohydrates to SCFA, which is the major energy source for the intestinal epithelial cells. Gut microbiota produces antimicrobial peptides to protect the host against pathogen colonization [51,52]. Gut microbiota stimulates the immune system by releasing ligands such as microbial-associated molecular patterns which bind to the host cell receptors. In a study conducted by Brisbin et al. [53], it was demonstrated that treatment of macrophages with *L. acidophilus* increased the expression of IFN-α. IFN-α activates the STAT2 pathway to increase MHC class I antigen processing and presentation and T-cell activation leading to the elimination of pathogens and prevention of dysbiosis.

In an ideal state, the beneficial bacteria compete with and prevent the colonization of pathogenic bacteria. An imbalance between beneficial and harmful intestinal microbiota results in dysbiosis [54]. Dysbiosis can be caused by non-infectious factors such as dietary changes, non-starch polysaccharides (NSPs), and mycotoxins or infectious agents such as *Clostridium perfringens* and *Coccidia* [55,56] which are summarized in Figure 2. Dysbiosis in broilers is characterized by intestinal inflammation and villus atrophy of the small intestine [57]. For many years, the use of antibiotic growth promoters (AGP) prevented dysbiosis in chickens. However, with the restrictions on the use of AGPs dysbiosis emerged as a challenge [58,59]. The most common factors that can cause dysbiosis are discussed below.

Stress can be of dietary, environmental, or managemental origin and can have detrimental effects on the health and production of poultry. During stress, the body redirects the nutrients towards inflammation which will decrease production [60,61]. Feed ingredients directly or indirectly influence the composition of gut microbiota, gut physiology, and gut-associated mucosal immune responses and can contribute to dysbiosis [62]. 

Non-starch polysaccharides (NSP) are naturally present in grains such as wheat, rye, and barley. NSP includes cellulose, glucans, inulin, arabinoxylans, chitin, gums, and mucilages and constitutes a major portion of dietary fiber [63,64]. Insoluble NSPs are resistant to digestion by mammalian enzymes and are fermented by the gut microbiome into SCFA [65]. However, soluble NSPs have negative effects on villus height, width and surface area, and gut immune response in poultry. Soluble NSPs increase the viscosity of gut digesta, increase the transit time and create an anaerobic environment in the gut [66,67] leading to colonization by pathogenic anaerobic microbes such as *C. perfringens* and *Escherichia coli* and acute inflammation [68,69].

Dietary proteins play a significant role in modulating the immune response. Amino acid composition of dietary protein modulates the immune system by altering the gut microbial composition [70]. Poultry feeds include plant protein such as soybean meal, and cottonseed meal, or animal protein such as fish meal [71]. A high proportion of animal proteins, which are rich in glycine and methionine, favors colonization by pathogenic *C. perfringens* and high levels of fish meal in poultry diets can lead to *C. perfringens* overgrowth and induce necrotic enteritis in chicken [72]. Undigested low-quality protein in the caeca will be fermented by caecal microbiota, in a process termed putrefaction, to produce toxic metabolites such as indole and cresol leading to inflammation and intestinal damage [73]. 

Heat stress is an environmental factor that negatively impacts chicken production performance and health. Heat stress decreases the expression of tight junction proteins resulting in leaky gut syndrome [74]. Heat stress can also cause immunosuppression, lipid peroxidation, and endocrine imbalances [75] leading to dysbiosis in chickens. In a study conducted by Park et al. [76], it was identified that heat stress increased the proliferation and colonization of enteric pathogens such as *Escherichia*, *Salmonella*, and total aerobic bacteria.

Mycotoxins are a diverse class of toxic secondary metabolites produced by fungi [77] and molds belonging to the genera *Aspergillus*, *Penicillium*, and *Fusarium* [78]. Because mycotoxins are present in the feed, the gut is the primary organ affected by mycotoxins [79]. The gut microbiome can metabolize mycotoxins, thereby affecting toxicity. Microbial enzymes are capable of chemical transformation of mycotoxins [80]. The toxicity of aflatoxin, a common mycotoxin, is dependent on the transformation by gut microbes. Mycotoxins can cause dysbiosis by their direct antimicrobial properties or by an indirect effect on intestinal cells. Mycotoxins cause apoptosis of the intestinal epithelial cells, increase intestinal permeability, increase mucus production, damage the intestinal villi, and disrupt tight junction proteins [81]. Mycotoxins increase the production of proinflammatory cytokines, immunoglobulin A, and antimicrobial peptides to alter the gut microbial composition [77,80].

Antibiotic growth promoters (AGPs) improve the performance and health of chickens by altering the gut microbiome. AGPs inhibit the gram-positive bacteria and pathogenic bacteria which compete with the host for nutrients [82] or produce toxins [83]. AGP reduces the total number and diversity of the gut bacteria [84], though the loss in gut microbial diversity due to the use of AGPs can also lead to dysbiosis in poultry. Gut infections can alter the balance in gut microbiota. In a study conducted on the effect of *Eimeria tenella* infection on the cecal microbiome diversity, it was identified that the diversity of cecal microbiota remained stable, but there were significant changes in the abundance of some microbial taxa. In birds with severe cecal pathology, there was a decrease in taxa belonging to the order *Bacillales* and *Lactobacillales* and an increase in Enterobacteriaceae. In asymptomatic birds, there was an increase in *Lactobacillus* and a decrease in *Bacteroides*. The variations in gut microbiota correlated with the *E. tenella* lesion severity [85].

## 8. Probiotics in Broiler Production

With the ban on the use of antibiotics by the European Union (EU) and the limited use of antibiotics in the United States (US) in chicken production, probiotics are emerging as a potential alternative. Probiotics or direct-fed microbials (DFM) are defined as “live microorganisms which when administered in adequate amounts confer a health benefit on the host”, by improving the gastrointestinal microbial balance and consequent enhancement in nutrient absorption, growth rate, feed efficiency, and immune competence against potential pathogens [84,86,87]. An ideal probiotic should be of host origin, have demonstrated a beneficial effect on the host, should be able to alter the gut microflora advantageous to the host, should be resistant to gastric acid and bile salts, should adhere to the gut mucosal epithelium, and should exhibit beneficial effects. Probiotics should also be non-pathogenic, free of adverse side effects, sensitive to antibiotics, exhibit antimicrobial properties against common pathogens, modulate intestinal functions, and immune responses of the host [88,89]. The probiotic should also be able to withstand the processing and storage conditions [90]. 

Ideally, probiotics should be host-derived because 1. The microbes have evolved and adapted to the ecology of the host gastrointestinal tract and hence, can readily proliferate and establish a stable population 2. Express beneficial effects better than probiotic strains obtained from other species [91]. It is therefore essential to develop host-specific probiotics for achieving optimal health and production. Probiotic bacteria should be typically isolated from the gut of healthy animals. Multiple species and strains of gut microbes must be evaluated for their probiotic potential and the one conferring maximum benefit to the host must be selected [92]. For example, only 6 out of 57 lactic acid bacteria (LAB) strains, isolated from the crop and intestine of healthy broilers, possess probiotic properties that included the ability to adhere to ileal epithelial cells, hydrophobicity, tolerance and survivability in gastric juice, bile salts, and phenol, susceptibility to antibiotics, ability to auto-aggregate and co-aggregate [93]. 

Poultry is a major source of foodborne pathogens of zoonotic significance [94] and foodborne pathogen infection cause economic loss to the poultry industry. Probiotics possess antimicrobial activity and can be applied to decrease the incidence of foodborne pathogens in poultry [95]. Probiotics secrete bacteriocins, hydrogen peroxide, alcohol, and organic acids and exhibit broad spectrum antimicrobial activity [96]. In a study conducted to evaluate the probiotic efficiency of LAB isolated from poultry, it was demonstrated that 77 strains out of the 296 strains screened inhibited the proliferation of *E. coli* and *Salmonella enteritidis*. Of the 77 strains, eight strains of their greater spectrum of pathogen antagonism and other desirable properties of probiotics. The strain with maximum probiotic potential was identified as *L. salivarius* [97].

The probiotic strains should be able to survive the acidic pH in the proventriculus and gizzard for the strains to successfully colonize the intestine [98]. The transit time of feed through the proventriculus and gizzard is approximately 2 h. *L. acidophilus* survives at pH 4 for 2 h [99]. pH tolerance is strain specific. For example, *L. plantarum* VJI21 exhibit 80% survivability whereas the isolate VJC1 exhibit only 64% survivability at low pH [100]. 

Potential probiotic strains should be able to withstand bile salts in the duodenum and cecum [101]. Probiotics of host origin will be able to tolerate the bile salt concentration of the host from which the probiotic strain was isolated. *L. acidophilus* can survive for 15 h in 2% bile. *Lactobacillus* strains possess bile salt hydrolase activity and therefore neutralize the antimicrobial activity of bile salts to survive in the gut [100]. 

The ability of probiotics to adhere to the intestinal epithelial cells and mucosal surfaces is essential for forming biofilms, which prevents the detachment of bacteria during gut contraction and peristaltic movement. This property of biofilm formation is termed auto-aggregation [102]. Co-aggregation enables the probiotic species to displace pathogenic bacteria [101]. Hydrophobicity enables probiotics to adhere to the gut epithelium [97]. Auto-aggregation, co-aggregation, and microbial hydrophobicity enables the probiotics to competitively exclude pathogenic species from the host gut [100,102,103]. *L. acidophilus* decreases the adhesion of *E. coli* and *S. typhimurium* to Caco-2 cells [99].

Antibiotic susceptibility of potential probiotics should be evaluated to ensure no antibiotic resistant genes are transferred from the probiotic species to the gut microbiota [100]. In a study evaluating antibiotic susceptibility of *Lactobacillus* strains isolated from chicken, 90% of the 88 isolates were resistant to tiamulin, 74% to tetracyclines, and 70% to lincomycin. Multidrug resistance was detected in 79.5% of isolates [104]. These results indicate that caution should be applied before choosing a particular probiotic species as the probiotic species might carry antibiotic resistant genes. 

Probiotic organisms should be consumed at adequate doses to confer a desired health benefit. A minimum viable cell concentration of 1 × 10^6^ CFU g^−1^ is essential for successful colonization of the gut by probiotics [105]. Probiotic supplementation at 1 × 10^8^ to 1 × 10^9^ CFU g^−1^ is effective and can increase the number of beneficial bacteria [90,106]. In a study conducted in broiler chickens by Mountzouris et al., 2010 [106], it was demonstrated that there is a significant increase in the concentration of *Lactobacillus* and *Bifidobacterium* and a decrease in coliform counts in the caeca of birds fed probiotics. Microbiological factors such as the strain of the probiotic, water activity, pH, presence of salt, and molecular oxygen determine the survivability of probiotics during storage. Feed processing parameters such as heat treatment, cooling rate, storage length, and packaging materials also determine the survivability of probiotics [107,108,109]. The survivability of probiotic bacteria during the processing and storage of the product is dependent on the species and strain of probiotics [110]. Thermotolerance of *Lactobacilli* can be increased by subjecting the bacterium to heat shock at sublethal temperatures, which is 10 °C above the optimum temperature for growth, before exposure to lethal temperatures [111]. *Bacillus* spores can survive at high temperatures and are stable during feed processing steps such as pelleting [105].

Probiotics can be a single strain or a combination of different microorganisms such as bacteria of the species *Lactobacillus*, *Bifidobacterium*, *Bacteroides*, *Bacillus*, or *Enterococcus*. *Saccharomyces* spp., a yeast, is used as a probiotic either alone or in combination with other bacterial strains [112]. Probiotic preparations can also contain multiple strains of the same species such as *Lactobacillus casei*, *L. acidophilus*, *L. bulgaricus*, *L. reuteri*, *L. salivarius*, and *L. animalis* [90,113,114]. Probiotics can also be categorized as ‘colonizing’ species which consists of *Lactobacillus*, *Streptococcus*, and *Enterococcus* spp., and ‘non-colonizing’ species which consists of *Bacillus* spp. spores and the yeast *Saccharomyces cerevisiae* [115]. Bacteria of the colonizing species compete for potential binding sites on the intestinal epithelium or mucosa while non-colonizing species though viable in the intestinal content, will not colonize the intestinal epithelium [116]. Colonizing species resist the growth of pathogens by competitive exclusion. In contrast to the requirement for continuous administration of non-colonizing species, colonizing species requires to be administered once, though a supplementary dosage at a later age can be beneficial [117]. Gut microbiota is also classified as luminal and mucosal associated microbiota. The expression of adhesion molecules on the intestinal epithelial cells, rate of mucus production, and secretory immunoglobulins determine the composition of mucosal microbiota [118]. The composition of luminal microbiota is influenced by the presence of antimicrobial substances, feed passage rate, nutrient availability, and the interaction between the microbiota and the host immune system [94,110].

Among the different probiotics that are available, three major microorganisms, namely Lactic acid bacteria, *Bifidobacteria*, and *Saccharomyces*, dominate the probiotic market [119]. Other probiotics that are available for the poultry industry include *Enterococcus*, *Pediococcus* [120], *Bacillus*, *E. coli*, *Aspergillus*, *Candida*, and several other microbial species [121]. Although the probiotics industry is rapidly growing with several commercially available probiotics flooding the market, at any given time it is customary to supplement birds with a combination of a limited number of four or five microbial species [122]. It is not clear how the supplemented four to five different strains can contribute to the bacterial diversity in healthy chickens or even restore diversity in antibiotic treated birds. In humans treated with antibiotics, gut mucosal microbiome reconstitution is impaired by probiotics suggesting that probiotics might prevent the original gut population from recovering to the pre-antibiotic phase [123].

One alternative to applying limited numbers of probiotic species would be to transfer gut microbiota from healthy adult chickens to young chicks. The poultry industry will benefit from transferring mixed cultures of gut microbiota from healthy chickens rather than supplementing a limited number of probiotic species. Such an attempt has been extensively applied to control *Clostridium difficile* infection in humans. Fecal microbiota transplantation is the transfer of fecal matter from healthy donors into the gut of recipients to alter the gut microbiome of the recipient [124]. Fecal microbiota transplantation increases gut microbiota diversity and restores the gut microbiome post-antibiotic therapy. Post-transplantation, the fecal microbiota of fecal transplant recipients was more diverse and similar to that of the healthy donor [125]. Such an approach was attempted in early 1973, wherein transferring an undefined mixed culture from healthy chickens, but not a single *Lactobacillus* probiotic species, successfully inhibited *Salmonella* colonization in chicks [126]. This procedure was termed the Nurmi competitive exclusion concept [121] and has since been applied to control salmonellosis and campylobacteriosis in poultry [127].

It is necessary to understand the symbiotic relationship between the host and the microbiota. A nutrient-rich ambient environment of the host facilitates the colonization and proliferation of microbiota while the microbiota stimulates the digestive and immune systems of the host to improve the production performance [89,128]. In a meta-analysis study conducted by Blajman et al. (2014) [129], it was identified that probiotic supplementation improves BWG by 661 g and improves FCR with 281 g less feed consumed/kg of weight gain. The meta-analysis study also concluded that administration of probiotics in drinking water is more efficacious compared to supplementation in feed. The major limitations impeding the use of probiotics are the inconsistent effects and the incomplete understanding of the mode of action of probiotics.

## 9. Routes of Administration

Probiotics should be administered as early in life as possible to achieve desired beneficial effects [130]. Supplementation in feed or water on the day of hatch is the most common and convenient method to apply probiotics in poultry production [86]. Supplementation in feed or water is dependent on the individual consumption and thus the dose of probiotic consumed varies depending on the quantity of feed or water consumed. Supplementing probiotics in feed and water provide a mean for continuous supplementation of the probiotics to the bird, though the presence of antimicrobials in food or water can affect the viability of the probiotic bacteria [131,132].

Several studies have reported higher efficiency of probiotics when supplemented in drinking water [129,133]. This can be due to the shorter transit time of water, compared to the feed, in the upper gastrointestinal tract. A shorter transit time will reduce the duration for which probiotics are exposed to lower pH and bile salts in the upper gut. Water dilutes the acid, enzymes, and other digestive secretions and thereby reduces their negative effects on probiotics [132]. Bacteria require a period of acclimatization to the new environment. This period of time, termed the lag phase, will be longer when the conditions are less optimal. Supplementing probiotics in water mimics the reconstitution of lyophilized products and reduces the lag phase, thereby enabling the bacteria to adapt better and proliferate in the gut [132,134,135].

Probiotics can be injected into the incubating egg to achieve early colonization of the beneficial bacteria [136]. Delivery of probiotics to chicken embryos in ovo establishes a healthy gut microbiome [137,138]. In ovo probiotic inoculation can be applied either as in ovo stimulation [139] or in ovo feeding [140] which are inoculation on day 12 and day 17–18 of egg incubation respectively. In in ovo *stimulation*, prebiotic species are deposited on the air cell on day 12 to stimulate the development of innate gut microflora, which is ingested when the chicks start piping [141]. In in ovo feeding, probiotic species are injected into the amnion or embryo [142,143,144]. In a study conducted by Oliveira et al. [136], it was demonstrated that in ovo inoculation of probiotics significantly reduced mortality when challenged with *Salmonella* on day-4 post-hatch compared to the control group. In ovo inoculation of probiotics facilitates early colonization of beneficial bacteria and competitively excludes *Salmonella*. However, in ovo injection of one or a mixture containing only a few beneficial bacteria will likely not be effective in establishing a diverse intestinal microbiome comparable to that of a healthy adult chicken. On the other hand, inoculation of a mixed culture can introduce unknown species of bacteria that could be detrimental to the embryonated egg [145].

Spraying is another method of probiotic administration wherein the undiluted culture is sprayed on chicks when they are 50–70% hatched [146]. Cultures can also be sprayed in the environment or the litter material of the bird. Spraying eliminates the water quality concerns and other variables associated with probiotic administration in feed or water. Spraying is a cost-effective and suitable for mass applications [147,148]. Probiotics can also be delivered by oral gavage, but the high labor requirement makes the route of delivery unfeasible [143].

## 10. Postulated Mode of Action and Effects of Probiotics

### 10.1. Competitive Exclusion and Antagonism

Competitive exclusion refers to the phenomenon in which a strain of bacteria competes and prevents the colonization of enteric epithelium by other bacteria [149]. Probiotics inhibit the growth of pathogenic bacteria and selectively enhance the growth of beneficial microorganisms through several mechanisms [150]. Probiotics produce organic acids and SCFAs such as lactate, propionate, acetate, and isovalerate to lower the pH of the gut and thus inhibit the colonization and growth of pathogenic microbes [151]. *Lactobacillus* and *Bacillus* spp. secrete bacteriocins to inhibit *E. coli*, *Vibrio* spp., *Salmonella*, *Proteus*, *Campylobacter*, *C. perfringens*, *Staphylococcus aureus*, and *Shigella* [152]. Our group earlier identified that the cell-free supernatant of P. acidilacti can inhibit the proliferation *C. jejuni* in vitro at 1:1 supernatant: pathogen dilution [153]. Probiotics also compete with pathogenic bacteria for nutrients and adherence to the gastrointestinal epithelium. We earlier found that probiotic supplementation during *Salmonella* challenge significantly reduced the cecal *Salmonella* load [154]. *Lactobacillus plantarum* hinders the adherence and translocation of *C. jejuni* to the enteric epithelium by inducing mucin secretion [155]. Probiotics can act as an adjuvant to stimulate the immune system, thus inhibiting pathogen colonization and reducing mortality [149,156]. 

### 10.2. Host Intestinal Health and Integrity

The three principal regions of probiotic colonization in the gut are the enterocytes, colonic and cecal epithelium [157]. Probiotics enhance the epithelial barrier integrity by increasing the production of mucus [158]. Probiotic bacteria enhance mucin secretion by the upregulation of MUC2 in goblet cells [159]. Probiotic bacteria regulate tight junction permeability through upregulating zonulin, a protein that regulates tight junction permeability [160]. In a study conducted in our laboratory, we demonstrated that supplementing *B. subtilis* to the chicken during necrotic enteritis increased the mRNA expression of claudin proteins 2.1-fold compared to the challenged control group [161]. *B. longum*, *Pediococcus pentosaceus*, *L. plantarum*, and *L. acidophilus* scavenge free radicals and thereby reduce host DNA damage and thereby improve gut integrity. Probiotics also reduce lipid peroxidation of the gut epithelial cells and maintain gut integrity [162].

The host immune system can distinguish pathogenic bacteria from probiotic bacteria [163]. It was demonstrated that the host produces IL-8 in response to pathogenic bacteria infection. Probiotic bacteria did not induce the production of IL-8 and reversed the pathogen-induced increase in IL-8 [164]. Probiotic bacteria induce the production of antimicrobial peptides and protective cytokines such as IL-12, IL-10, TNF-α, and IFN-γ through their interaction with the Toll-like receptors (TLR) on epithelial cells. This enhances the regeneration of intestinal epithelial cells and decreases apoptosis in intestinal cells by inhibiting the pro-apoptotic pathway (MAPK/p38) and activating the anti-apoptotic pathway [160,165]. In a study conducted by Gharib et al. [166], it was demonstrated that Bacillus amyloliquefaciens supplementation downregulates IFN-γ expression and upregulates IgG and IgM. Bacillus amyloliquefaciens supplementation decreased caspase-3 and enhanced occludin expression to maintain gut integrity [166].

### 10.3. Digestion and Absorption

Probiotics increase the nutritional value of the feed by fermenting undigestible carbohydrates. The fermentation of undigested carbohydrates and proteins in the ileum and colon by probiotics produces SCFAs, which provide energy to the host [167]. DFM supplementation increased the epithelial flux of D-glucose, DL-methionine, and L-lysine and increased trans-epithelial resistance, which is an indicator of the tight junction integrity [168]. Probiotic supplementation increases the ratio of villus height to crypt depth resulting in increased intestinal absorption of nutrients [169]. We earlier identified that probiotic supplementation during *Salmonella* challenge in broilers reversed the loss in villus height and crypt depth [154]. The effect of probiotics on villus height and crypt depth can be correlated with the increased FCR and BWG in the probiotic supplemented birds. Some probiotic bacteria such as Bifidobacterium and *Lactobacillus* spp. synthesize vitamins such as vitamin A, vitamin K, folate, nicotinic acid, pyridoxine, riboflavin, thiamine, cobalamin, pantothenic acid, biotin, and increase the activity of antioxidant enzymes such as glutathione peroxidase, catalase, superoxide dismutase, and glutathione S-transferase which adds to the nutritional value of the probiotic [170,171,172]. The exopolysaccharides (EPS) produced by probiotic bacteria aid in biofilm formation and modulate the immune system [173]. Probiotics secrete extracellular enzymes such as amylases, phytases, proteases, and lipases to increase feed digestibility [174]. 

### 10.4. Performance Parameters

Probiotics enhance the health of birds and consequently improve production performance. Probiotics improve feed intake by decreasing gastric emptying time [175]. The increased weight gain can be attributed to better feed digestibility associated with the secretion of the enzymes amylase, lipase, and protease to improve nutrient availability [176]. Probiotic supplementation improves the live weight gain, carcass yield, and cut-up parameters in broilers [177]. Probiotic supplementation improves the microbiological quality of meat by reversing the pathogen-induced loss in gut permeability [121]. Probiotics are capable of modulating bone health by regulating the absorption of calcium and phosphorus in the intestine and secretion of neuroendocrine molecules (incretins and serotonin). Gut dysbiosis leads to decreased bone density and strength [178,179]. Yan et al., 2019 [180] demonstrated that probiotic supplementation increased mineralization of the femur and tibia resulting in significantly higher bone strength and gait score. Nevertheless, some studies have reported decreased growth and performance parameters associated with the supplementation of probiotics and symbiotics [181,182]. Several factors such as the probiotic strain, dosage, general health of the bird, sex of the bird, duration of poultry house downtime, and environmental contamination levels determine the success of probiotic application in poultry [183].

Probiotic supplementation mitigates heat stress by restoring the intestinal microbial balance and preventing dysbiosis associated with heat stress [180]. Heat stress decreases ileal and caecal Bifidobacterium and *Lactobacillus* loads and increases coliforms and Clostridium loads, which was reversed by probiotic supplementation [184]. Probiotics decrease the expression of heat shock proteins and decrease the adhesion sites for pathogenic bacteria and thereby decreasing pathogenic bacteria colonization [185].

### 10.5. Immunomodulation

The gastrointestinal system is considered to be the largest immune organ. The gut-associated lymphoid tissue (GALT) is constantly being exposed to environmental and dietary antigens [158]. The innate immune system is usually the first line of defense against pathogenic intestinal microbiota [13]. Mucins and antimicrobial peptides such as defensins and lysozyme are secreted by epithelial cells in response to cues from the intestinal microbiota. Mucins cover the epithelium and regulate the diffusion of macromolecules, nutrients, gases, and toxins while serving as an adhesion site for resident microbiota [186,187]. Probiotic bacteria produce SCFAs which stimulate the expression of mucin glycoprotein. Butyrate is utilized as an energy source by the colon epithelial cells, improves gastrointestinal function, and exerts anti-inflammatory effects by inhibiting NFκB [188]. In a study conducted by Zhou et al. [189], it was demonstrated that butyrate supplementation in poultry significantly decrease nitric oxide production by macrophages in a concentration dependent manner by possibly inhibiting the activation of NFκB. Butyrate also inhibits the expression of pro-inflammatory cytokines IFN-γ, IL-6, and IL-1β in LPS-activated macrophages. 

Intestinal epithelial cells and intraepithelial lymphocytes express TLRs which are PRRs and can be classified as the ‘gate keepers’ of the innate immune system. The binding of ligand to the PRRs initiates a cascade of signaling pathways that include interferon response factors, mitogen activated protein kinases, NFκB, and activator protein 1 leading to the secretion of proinflammatory cytokines [190]. TLR signaling plays an important role in maintaining the epithelial barrier function, intestinal epithelial homeostasis, and epithelial cell proliferation. In the enterocytes, TLRs are expressed on the basolateral surface to prevent the interaction of TLRs with commensal microbiota. However, TLRs are also expressed on the luminal surface to a lesser extent which plays an important role in gut homeostasis [191]. Stimulation of the TLRs by the normal gut microbiota induces the production of IL-6 and KC-1 which are tissue protective factors [192]. Pathogens equipped with virulence factors traverse the epithelium, resulting in the activation of TLRs located in IEC, macrophages, and dendritic cells resulting in inflammation [193]. Probiotic bacteria preferentially activate the TLR9 pathway which attenuates NFκB activation and IL-8 expression. Probiotic bacteria activate the TLR9 signaling pathway and upregulated the anti-inflammatory cytokines in mice [194]. This study suggests the significance of TLR signaling by probiotic bacteria in maintaining gut homeostasis.

Probiotics increase the expression of TLRs, increase antibody production, and increase the number of intraepithelial lymphocytes in the gut [195]. Probiotics induce the secretion of IgA in mucins to inhibit pathogen colonization [196]. Probiotics can enhance macrophage activity, induce specific and non-specific humoral and cellular immune responses, enhance cytokine production and secretion of antibodies (IgA, IgG, and IgM) [170]. In a study conducted by Bai et al. [197] probiotic supplementation significantly increases the CD4+, CD8+, CD3+, and intraepithelial lymphocyte populations. Probiotics supplementation increases the expression of TLR 2 and TLR 4. Probiotics maintain the balance between the pro-inflammatory and anti-inflammatory cytokines [198,199]. Probiotics shift T-cell maturation towards the Th1 pathway through the modulation of cytokines [156]. In contrast to pathogenic bacteria, which activate the proinflammatory transcription factor NFκB, probiotic bacteria activate the regulatory factor IκB thus exerting anti-inflammatory effects. Inhibition of the NFκB-pathway and the increased production of cytoprotective heat shock proteins by probiotics confer an advantage during heat stress [200].

## 11. Factors to Be Considered during Probiotics Supplementation

The survivability of the probiotic organisms in the upper gastrointestinal tract is of paramount importance for the probiotic species to colonize the gut. Microencapsulation preserves the viability of probiotics in a feed matrix. However, in vitro tests with simulated gastric juices are typically applied to study the efficiency of microencapsulation in maintaining the viability of probiotics [201]. The high complexity of the gastrointestinal tract of live animals with variable pH, gastric and intestinal juices, peristalsis, and presence of gut microbiota makes in vitro studies to be of low predictive value for in vivo success. *L. casei* failed to colonize mice’s gut when supplemented with sodium caseinate capsules [202]. More studies have to be conducted using in vivo chicken models to assess the protective effect of microencapsulation on probiotic survival.

Even if the commercial probiotics remain viable on reaching the poultry gut, direct evidence to show the colonization of probiotic species post supplementation is lacking. The survival and colonization of a probiotic species are a function of several factors including bile salts and pH. Considering that the pH of the chicken intestine ranges from 3.1 to 6.6 [203], it is possible that the supplemented probiotics may even fail to colonize the gut. The classic approach to identifying the survival and colonization of the in-feed probiotic species is to isolate them by cultivation in species-specific agar plates [204]. But this technique has limitations in that the supplemented probiotic species have no antibiotic resistance genes and species-specific media is not available for most of the probiotic species. Advanced genome-based techniques though have been extensively used to determine the diversity and structure of the microbial population, they cannot quantify individual bacterial species. Because of these constraints, very rarely attempts have been made to quantify the survival and colonization of the supplemented probiotics. A recent article from our group identified that out of the five supplemented probiotics, only four species successfully colonized the gut post supplementation and were identified at levels ranging between 0.03 and 1.2% depending on the region of the intestine and the probiotic species studied [205]. Among the sparse articles that measured the survival of the supplemented probiotics in the poultry gut, it is not clear whether it can be stated with certainty that the probiotics, in fact, colonized the gut considering the fact that humans supplemented with probiotics only transiently increased the probiotic species concentration in their feces [206]. 

One of the major factors that can prevent the colonization of the gut by the supplemented probiotics is “Colonization resistance”. Colonization resistance is loosely defined as a mechanism by which the resident microbiota prevents colonization of exogenous microorganisms [207] where the existing microorganism prevents the probiotics from colonizing the gut. In mice colonization of the gut was limited by the colonization resistance of the existing microbiome [206]. Humans can be classified as “permissive” or “resistant” to the colonization of a particular strain of probiotics based on their genetic makeup [208] and the probiotic colonization pattern of individual humans can be predicted using the microbiome features of the individual [206]. The poultry industry might benefit from probiotics that are designed for specific breeds of birds so that the bacterial strains in the probiotics have a greater chance of colonization.

Another extension of these individualized breed-specific probiotics would be developing probiotics specific for age. Although humans [209], mice [210], and poultry [211] show age-specific differences in their gut microbiota composition, it is common to find that the same combination of four to five probiotic species is being fed to birds of all ages to counter all disease conditions. Developing age-based probiotic supplements should be one of the future approaches for developing new probiotics for the poultry industry.

Most probiotics are a normal inhabitants of the host gastrointestinal tract. The presence of antimicrobial resistance genes in bacteria such as Bifidobacteria and Lactobacilli by itself is not a matter of concern because of their lack of infectivity. Antibiotic resistance in probiotic bacteria is desirable to some extent as it can restore the gut microbiota after antibiotic therapy [212]. But the abundance of these bacteria in the gut presents the risk of transfer of antibiotic resistance genes from these probiotic species to pathogenic bacteria [213]. Transfer of these genes from animals or meat to humans through feces, soil, water, or food can result in the development of antibiotic resistance in human pathogens and treatment failure during infection [214]. Consumption of raw or undercooked meat can also result in the transfer of antibiotic resistance genes harboring zoonotic pathogens such as *Salmonella* and Campylobacter from poultry [215,216]. 

Generally, probiotic supplementation is associated with improved performance parameters including enhanced weight gain, superior carcass quality, increased carcass yield, improved meat sensory characteristics, and increased egg size, shell strength, shell thickness, and weight [217,218,219]. A commonly used bacterial species in probiotics is *Lactobacillus* spp., which is known to reduce gastrointestinal pathogen load, stimulate the immune system, and improve the growth and performance of the bird. However, direct exposure of rooster semen to *Lactobacillus* acidophilus was observed to be detrimental to semen quality. In a study conducted by Haines et al. [220], it was observed that exposure of rooster semen to *Lactobacillus* and Bifidobacterium virtually rendered the sperm immotile. The pH of semen was also found to be significantly reduced on exposure to these bacteria. This could be due to the direct attachment of the bacteria to the sperm, obstructing the movement of sperm by the non-motile bacteria, or reduction in pH due to the production of lactic acid. Therefore, long-term supplementation of *Lactobacillus* and Bifidobacterium containing probiotics to breeder stock should be done with caution.

## 12. Postbiotics and Paraprobiotics

The efficacy of probiotics is associated with the viability of the microorganisms used. However, there is evidence that the viability of the microbes is not a requisite for conferring the beneficial effects of probiotics to the host [221]. The emergence of terms such as ‘postbiotics’, ‘paraprobiotics’, ‘metabiotics’, ‘inactivated probiotics’, and ‘ghost probiotics’ indicates that supplementing non-viable microbes or microbial products can also confer health benefits to the host. Postbiotics are low molecular weight soluble factors that are either secreted by live bacteria or released after bacterial cell lysis and when administered in sufficient amounts confer health benefits to the host [221]. Postbiotics are obtained by disrupting the microbial cell structure through heat treatment [222], enzymatic treatment [223], solvent extraction [224], or sonication [225]. Postbiotics are further purified through centrifugation, column purification, freeze-drying, microfiltration, and dialysis [226,227].

Paraprobiotics also termed as “inactivated probiotics” or “ghost probiotics”, are non-viable probiotic or non-probiotic microbial cells or cell fractions which when administered in sufficient amounts confer beneficial effects to the host [228,229]. Paraprobiotics are essentially non-culturable, but immunologically active microbial cells that benefit the host. Thermal treatment, high pressure, irradiation, and sonication, which induce cell death without membrane degradation, are applied in the production of paraprobiotics [228,229,230,231,232]. Flow cytometry using fluorescent dyes is used to assess the functional state of inactivated cells [233].

Postbiotics (Nonbiotics) are low molecular weight non-viable factors such as metabolic products, byproducts, or cell wall components (metabiotics, cell-free supernatants, secretions, cell lysates, or biogenic metabolites) derived from probiotic microorganisms which when administered in sufficient amounts confer health benefits to the consumer [221,229]. Postbiotics can be soluble substances secreted by the live bacteria or the products of bacterial cell lysis, as summarized in Figure 3. This soluble fraction contains SCFAs, bacteriocins, vitamins, peptides, organic acids, hydrogen peroxide, enzymes, cell surface proteins, plasmalogens, peptidoglycan-derived muropeptides, teichoic acids, exopolysaccharides and endopolysaccharides [229,234,235]. The exact mechanism of action of postbiotics is not completely elucidated. In a study conducted by Yan et al. [236], it was demonstrated that *Lactobacillus* rhamnosus derived protein p40 activates EGFR and prevents the cytokine-induced apoptosis of intestinal epithelial cells in inflammatory conditions such as inflammatory bowel disease (IBD). In a study conducted by Humam et al. [237], it was demonstrated that the birds fed postbiotics derived from different strains of L. plantarum exhibited significantly improved FCR values compared to the control group. Postbiotic supplementation increased the expression of IGF-1 and growth hormone receptors during heat stress [237].

Although the mechanism of postbiotic action is not completely explained, scientific data provide evidence for the role of postbiotics in several physiological functions. Postbiotics can be classified based on their structural composition or their physiological functions. Based on their role in host physiology, postbiotics can be 1. antimicrobials such as cell-free supernatants of *L. plantarum* [238], 2. immunomodulatory substances such as cell wall components of Bifidobacterium and *Lactobacillus*, lipoteichoic acid of *L. plantarum* [224], 3. anti-inflammatory factors such as cell free supernatant of *L. paracasei* and *L. rhamnosus* [236,239], 4. anti-proliferative substances such as sonicated cell suspension of *L. casei* [240], 5. anti-obesogenic compounds such as fragmented cells of *L. amylovorus* [241], 6. Hypocholesterolemic substances [231], 7. anti-hypertensives such as polysaccharide glycopeptide complexes of *L. casei* [227], and 8. antioxidants such as intracellular contents of *Streptococcus salivarius*, *L. acidophilus*, and *L. casei* [236,240]. Supplementation of bacteriocin from *L. plantarum* improves growth rate, increases the fecal LAB population, and reduces the abundance of Enterobacteriaceae in broilers [242]. Thus, the concept of feeding live microbials is being replaced by postbiotics which have been demonstrated to confer similar health benefits to the host.

Based on the composition, postbiotics can be 1. carbohydrates such as galactose-rich polysaccharides and teichoic acids 2. lipids such as propionate and butyrate 3. proteins such as p40 molecule and lactocepins 4. organic acids such as propionic acid and 3-phenyl lactic acid 5. vitamins such as B-complex vitamins 6. complex molecules such as lipoteichoic acids and peptidoglycan derived muropeptides [234,243,244]. In a study conducted by Abd El-Ghany et al. [245], it was reported that supplementation of inactivated *Lactobacillus* to chickens improved the immune response to Newcastle disease virus vaccines. The improved immune response of the birds is due to the lipopolysaccharides, lipoteichoic acid, and teichoic acid present in the bacterial cell wall which acts as adjuvants to stimulate the host immune response. Lysis of the inactivated bacteria in the host gut releases nuclear antigens which might stimulate the host humoral and cell-mediated immune responses.

One of the advantages of postbiotics and paraprobiotics over probiotics is the possibility of postbiotics and paraprobiotics applications in immunocompromised patients. Administration of live microbes presents a risk to immunocompromised patients because of the possibility of occurrence of opportunistic infections and metastasis of probiotic microbes. For instance, translocation of the Streptococcus gallolyticus from the gut to the bloodstream has been implicated in colorectal cancer in humans [246]. Paraprobiotics or postbiotics such as SCFA have antimutagenic activities and selectively target cancer cells and can be applied to treat cancer patients [244]. Postbiotics have a clear chemical structure, longer shelf life, and safety dose parameters. In a study conducted by Shigwedha et al. [247], it was observed that probiotic cell fragments, the structural components of probiotic cell lysates, exert a broad spectrum of immunomodulatory functions. The shelf stability of probiotic cell fragments can be up to 5 years. The use of postbiotics can also eliminate the risk of prevalence and transfer of antibiotic resistance genes from probiotic species to the consumer [248].

## 13. Future Prospects: Designer Probiotics and Postbiotics

The concept of designer probiotics, where probiotic strains are genetically modified to improve gut survival and resistance to stress to enhance therapeutic effects has been attempted in humans [249]. For example, Bifidobacterium breve that was modified to increase the expression of BetL, a betaine uptake system, had a considerably improved osmotolerance, survival in simulated gastric juice, and growth rate [250]. Several such modifications have been attempted to improve the quality of human life. Genetically engineered *Lactobacillus* reuteri modified to express phenylalanine ammonia lyase enzyme have been applied to treat phenylketonuria [251].

Although designing probiotics to treat chicken genetic diseases can be an overstretch, the development of designer probiotics to treat or prevent microbial diseases or control foodborne pathogens is a real possibility in near future. Recombinant *Lactococcus lactis* that secrete antimicrobial peptides such as A3APO and Alyteserin inhibited the growth of pathogenic *E. coli* and *Salmonella* by 20-fold [252]. Supplementing Lactococcus lactis, modified to express the VP8 spike protein of the human rotavirus, provided 100% protection against rotavirus infection in a murine model [253]. Similarly, probiotics have been successfully designed to sequester toxins produced by Shiga toxigenic *E. coli*, *Vibrio cholerae*, *Clostridium difficile*, and *Listeria* in human and mice models of infection [254,255]. Considering the economic burden caused by *Salmonella* and *E. coli* to the poultry industry and since no single solution exists to control any foodborne pathogen of poultry origin including *Salmonella* and *E. coli*, a logical future step would be to develop genetically modified probiotics to control foodborne pathogens. Also, supplementing probiotics capable of producing antimicrobial peptides can overcome the limitations such as short half-life, production costs, and purification costs associated with the production of antimicrobial peptides.

Probiotics can be modified to deliver immunogenic molecules and antigens to the mucosal surface and hence can act as vaccines. Oral vaccination of mice with *L. casei*, modified to express the enterotoxigenic Escherichia coli fimbrial antigens protected more than 80% of mice challenged with a lethal dose of enterotoxigenic *E. coli* [254]. L. lactis engineered to express rotavirus spike protein VP8 induces anti-VP8 IgA in mice [253]. There are several advantages of utilizing recombinant probiotics as a vaccine vector. Antigens delivered orally through recombinant probiotics will be resistant to degradation by proteases in the gut. Further, recombinant probiotics are easy to administer and mimic the route of infection [256]. Hence, recombinant vaccine vectors using probiotic species is a future approach with the potential to control foodborne pathogen infections in poultry.

With a more thorough understanding of the mechanism of action of postbiotics, designer postbiotics can be used for targeted prevention and control of enteric diseases in poultry. In humans, supplementation of SCFAs and tryptophan was found to have a therapeutic effect on IBD [257]. In a study conducted by Tsilingiri et al. [239], it was found that potent postbiotics can prevent the colonization of *Salmonella* on healthy intestinal epithelium and downregulate pro-inflammatory pathways in IBD on a polarized ex-vivo organ culture model. Although extensive research on postbiotics is undertaken for various human disease models, only a few studies have been conducted on poultry disease models. In a study conducted by Chaney et al. [258], it was demonstrated that supplementing postbiotics (*Saccharomyces cerevisiae* fermentation product) in broiler diet significantly reduced the cecal load and prevalence of *Salmonella* enterica compared to the group fed a standard diet. The large-scale production of postbiotics and associated cost, considering the low yield of metabolites from cells, is limiting their widespread commercial application in the poultry industry.

## 14. Prebiotics

Prebiotics are defined as “selectively fermented ingredients that result in specific changes in the composition and/or activity of the gastrointestinal microbiota, thus conferring benefits upon host health” [259]. Compounds of carbohydrates considered prebiotics include fructo-oligosaccharides (FOS), galactooligosaccharides (GOS), Mannanoligosaccharide, oligochitosan, inulin, pyrodextrins, stachyose, maltooligosaccharides, isomaltooligosaccharide (IOS), xylooligosaccharide, glucooligosaccharides, soya-oligosaccharide, lactitol, lactulose, pectin, and arabinoxylan. Non-carbohydrate compounds such as polyphenols, certain lipids, peptides, and proteins are also considered candidate prebiotics. These macromolecules are either synthesized by microorganisms or obtained from plants [90,260,261,262,263]. However, the beneficial effects of prebiotic supplementation depend on the individual components. The chemical structure (short-chain or long-chain), duration of ingestion, and dose of the prebiotic determine the effects of prebiotics on gut microbial composition [262,264]. The desirable characteristics of a prebiotic are 1. Resistance to gastric pH, hydrolysis by host enzymes, and absorption in the gut 2. Fermentable by the intestinal microflora 3. Selective stimulation of intestinal bacteria that positively contribute to host health and wellbeing 4. Enhance local and systemic immunity against pathogen invasion [82,90].

Some of the advantages of prebiotic supplementation include the reduced occurrence of enteric diseases, improved gut health, improved performance, decreased odor, and enhanced nutrient utilization [263,265]. The mechanism of action of probiotics is 1. Positive modulation of intestinal microbiota for host benefit 2. Improved gut epithelial health 3. Immune system stimulation [266]. Prebiotics prevent the colonization of the intestinal epithelium by pathogenic bacteria directly by adsorption or by competitive exclusion wherein prebiotics preferentially promote the growth of beneficial microorganisms [267]. Prebiotic supplementation significantly reduces Campylobacter spp. loads by 1.0 log_10_ CFU g^−1^ in the caeca by inhibiting the adhesion of Campylobacter mannose-binding lectins to the host enterocytes [268,269]. The metabolization of prebiotics by the gut microorganisms results in the production of lactic acid, SCFAs, or antibacterial peptides such as bacteriocins which can modulate the composition and/or activity of the gut microbiota, conferring health benefits to the host as summarized in Figure 4. The SCFA produced by the fermentation of prebiotics by bacteria acts as an energy source for the intestinal epithelial cells, thus promoting gut health and integrity [270]. Prebiotics are non-pathogenic antigens capable of stimulating the host immune system beneficially [266]. In a study conducted by Janardhana et al. [271], it was demonstrated that prebiotics affects the systemic antibody levels and proliferation of caecal tonsil immune cells. These effects are mediated by the modulation of gut microbiota by prebiotics. Prebiotics selectively enrich the microorganisms such as *Lactobacillus* spp., and Bifidobacteria spp. which are able to utilize the non-digestible, but fermentable substrates [262]. Further, prebiotics inhibits the attachment of pathogens such as *Salmonella* [272], *E. coli* [273], and *Campylobacter* [274] to the receptors on host intestinal cells by acting as decoy receptors [272], and thus in addition to contributing to better growth and performance, it also improves the microbiological quality and safety of poultry products. 

## 15. Synbiotics

A product containing a combination of prebiotic and probiotic is termed symbiotic. The prebiotic compound in the combination selectively promotes the growth of the probiotic by providing the specific substrate for fermentation [275,276]. A summary of the concept is illustrated in Figure 5. In a study conducted by Mookiah et al. [277] on the effects of probiotics, prebiotics, and synbiotics in broiler performance parameters, significant improvement in weight gain and feed conversion rate were observed. In addition, there was a shift in caecal microbiota with an increase in the population of lactobacilli and bifidobacteria and a significant decrease in *E. coli* at day 21 of age. Increased caecal VFA production was also noticed in the groups fed probiotics prebiotics and synbiotics. However, the findings did not suggest the synergistic effect of probiotic and prebiotic as the synbiotic group performance parameters were not significantly different from the groups fed probiotic or prebiotic alone [277]. The results of the study conducted by our group are in agreement with the above study where we identified that the FCR and BWG were not different between the birds supplemented with Bacillus subtilis probiotic and the mannooligosaccharide symbiotic group [161]. Further, In a study conducted by Chen et al. [278], it was reported that the effects of synbiotic supplementation were comparable to that of AGPs. Synbiotic supplementation improved the feed conversion efficiency, decreased abdominal fat yield, and decreased drip loss in breast muscle in ducks. These effects are due to the modulation of intestinal microbiota, maintenance of intestinal barrier integrity, positive modulation of the immune system, and improved lipid metabolism by the components of synbiotics. There is also evidence of increased antibody production and immune response when the symbiotic fed groups are vaccinated. The positive effects of synbiotics are dose-dependent. Adverse effects such as decreased villus surface area in the jejunum and ileum were observed when the dose was doubled from 0.1% to 0.2% synbiotic supplementation [279].

## 16. Conclusions

The gastrointestinal microbial community plays an important role in many physiological and immunological systems. In poultry, the gastrointestinal tract is sterile at hatch, providing the poultry industry with a unique opportunity to have significant control over microbial colonization in young birds. However, due to increased variability in gut microbiota between birds based on breed, housing, environment, diet, and immune status, a clear relationship between host and microbiota is not established. Prebiotics, probiotics, postbiotics, and synbiotics have been studied in depth over recent years in response to increased limitations on the use of antimicrobials in poultry. These additives can alter chicken gut microbiota positively and thereby improving growth, performance, and immune parameters. Designer probiotics customized for specific breeds, age groups, and diseases have the potential to completely replace antibiotics in poultry. With the development of appropriate technology, gut microbiota can be routinely monitored and modulated for host benefit in near future.

## Figures and Tables

**Figure 1 foods-11-01401-f001:**
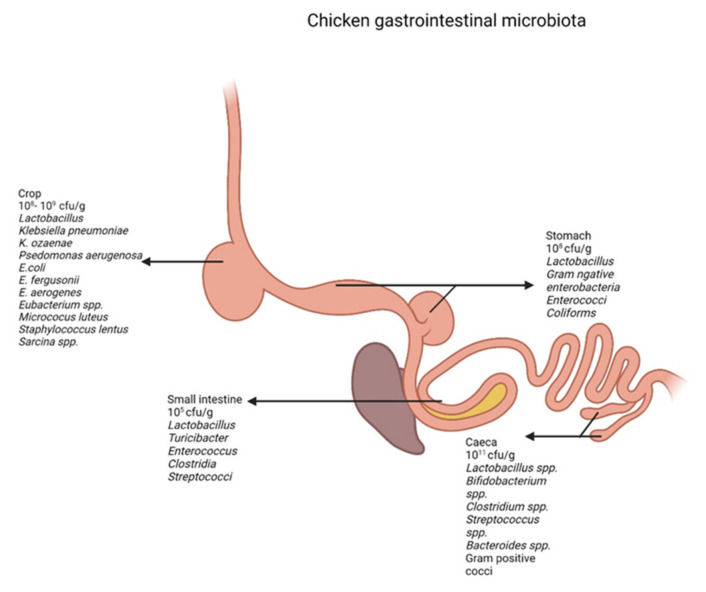
Regional abundance and diversity of gastrointestinal microbiota of chicken. Created with BioRender.com (26 March 2022).

**Figure 2 foods-11-01401-f002:**
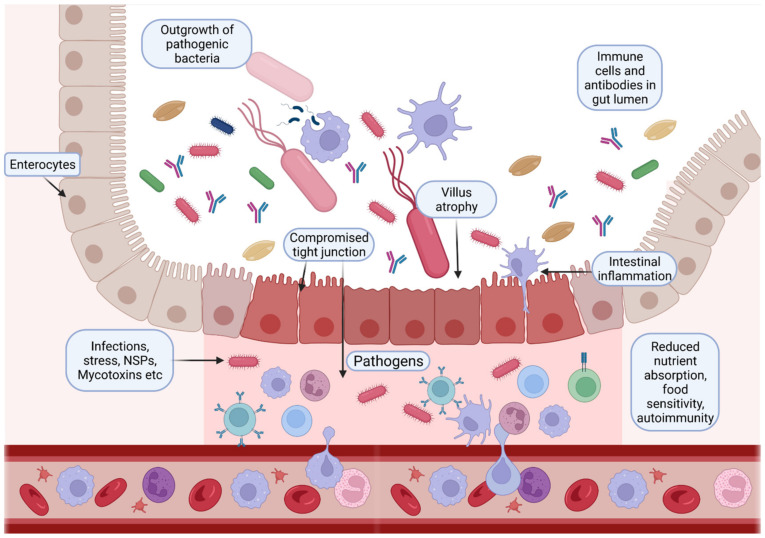
Dysbiosis induced by different factors alters the gastrointestinal homeostasis causing impaired epithelial barrier function and systemic inflammation. Created with BioRender.com (26 March 2022).

**Figure 3 foods-11-01401-f003:**
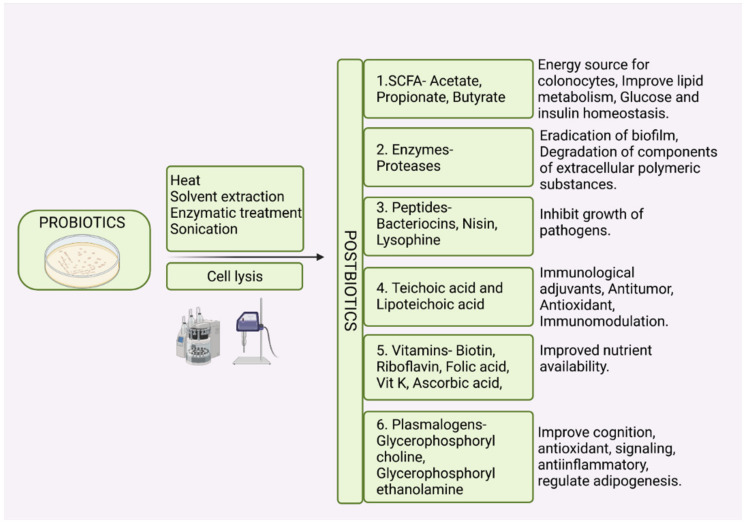
Postbiotics are soluble low molecular weight metabolites or cell lysis products derived from live or inactivated probiotic bacteria which when administered in adequate quantities demonstrate beneficial effects on host health. Created with Biorender.com (4 April 2022).

**Figure 4 foods-11-01401-f004:**
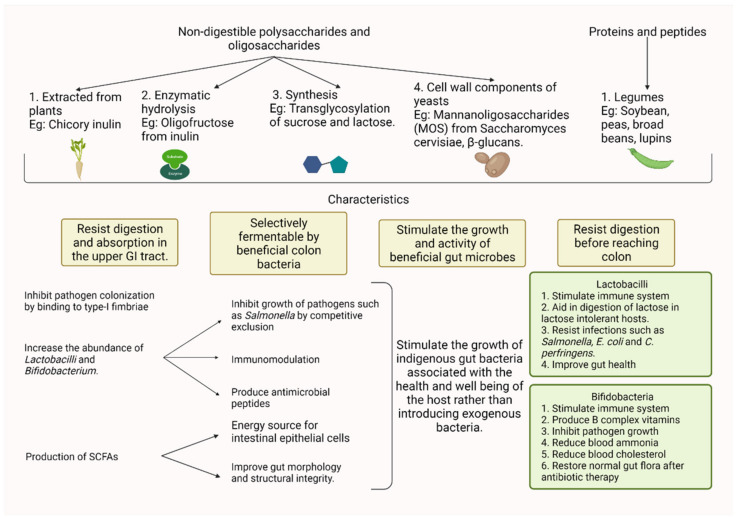
Prebiotics are polysaccharides or oligosaccharides capable of resisting digestion and absorption in the proximal intestine and is selectively fermented by caecal and colonic bacteria such as *Lactobacillus* and *Bifidobacterium*, thus increasing their abundance in host gut. Prebiotics act as decoy receptors for the binding of pathogens, thus preventing their attachment to the host intestinal cells. Prebiotics also serve as a substrate for the production of SCFAs which serve as energy source for the intestinal epithelial cells. Created with Biorender.com (26 April 2022).

**Figure 5 foods-11-01401-f005:**
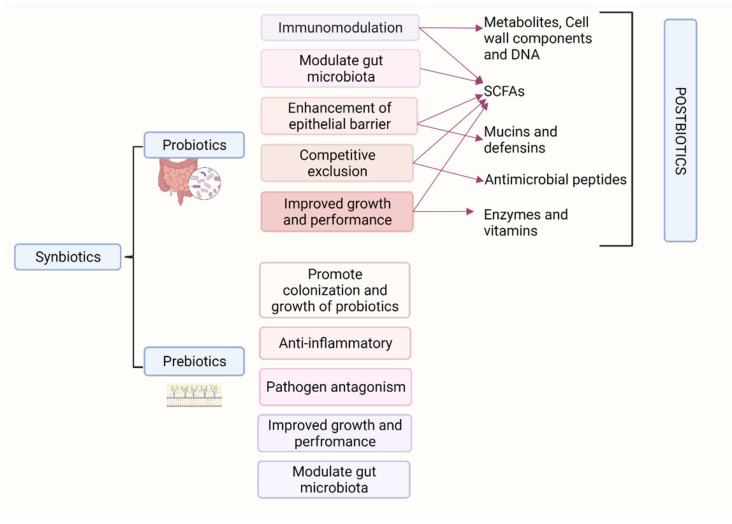
Synbiotics are essentially a combination of probiotics and prebiotics where the probiotic component is specifically fermentable by the prebiotic component and thus helps in establishing a stable population in the host gut. Created with Biorender.com (28 April 2022).

## Data Availability

No new data were created or analyzed in this study. Data sharing is not applicable to this article.

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
