# Peer review of "Gastrointestinal Microbiota and Their Manipulation for Improved Growth and Performance in Chickens"

_foods, 2022, doi:10.3390/foods11101401_

Round 1
Reviewer 1 Report
This review needs deep English revision by the authors and the authors should follow the guidelines of the journal. In addition, two parts (prebiotic and synbiotic), the authors superficially written these two parts and should be revised to improve the quality of this review
L16. Please delete factors
L23-27. Please rephrase, ''chicken'' repeated 3 times
L44. In vitro should be italic
L139. Walugembe, et al. [39], please add the number of ref. beside the authors' name. Please correct throughout the manuscript.
L143. Wielen et al. [41]
L145-165. Define the abbreviations at first mention
L153. Add ref. number
L376. Mountzouris et al., 2010, add ref number
L411. Blajman et al. (2014), please revise
L475. short chain fatty acids (SCFA), the abbreviation was mentioned earlier
L478. Replace like by such as or for example. Please revise throughout the manuscript
L472-482. Please revise names of bacterial strain, should be italic. Please check throughout the manuscript
L541. Performance parameters, the authors should focus on these parameters in more details. It discussed superficially
L579. This paragraph should focus on the mode of action regarding the effect of probiotic on TLR and NFkB
The effect of probiotic on immune organs, gut mucosal immunity and gut histology should be also considered by the authors.
L735. Why the authors added the mechanism of probiotic here, please delete
L740. SCFA has also immunity effect on chickens. Please define.
In fact, the prebiotic and synbiotic parts in this review are superficially written and needs revision by the authors
Why figures are at the end of the review please put it within the text
Author Response
Language and grammar check done using Grammarly.
Prebiotic and synbiotic sections revised. Extensive correction was made througjout the document. Please see the version with track changes
L16. delete factors
Corrected as suggested.
L23-27. Please rephrase, ''chicken'' repeated 3 times
Corrected as suggested
In vitro should be italic – revised throughout the literature.
Corrected as suggested
L139. Add the number of ref. beside the authors' name. Please correct throughout the manuscript – corrected throughout the literature.
Corrected as suggested
L475. short chain fatty acids (SCFA), the abbreviation was mentioned earlier - corrected
L478. Replace like by such as or for example- revised
L579. This paragraph should focus on the mode of action regarding the effect of probiotic on TLR and NFkB- included in the paragraph from 708 to 749
L740. SCFA has also immunity effect on chickens. Please define.
Probiotic bacteria produce SCFAs which stimulate the expression of mucin glycoproteins. Butyrate is utilized as an energy source by the colon epithelial cells, improves gastrointestinal function, and exerts anti-inflammatory effects by inhibiting NFkB [197]. In a study conducted by Zhou et al., [198][195] it was demonstrated that butyrate supplementation in poultry significantly decreased nitric oxide production by macrophages in a concentration dependent manner by possibly inhibiting the activation of NFkB. Butyrate also inhibited the expression of pro-inflammatory cytokines IFN-γ, IL-6, and IL-1β in LPS-activated macrophages.
Why figures are at the end of the review please put it within the text – Included within the text.
Corrected as suggested
Reviewer 2 Report
The review article cover many different topics to address “Gastrointestinal Microbiota and their Manipulation for Improved Growth and Performance in Chickens. Even within the last year, there have been over 10 review articles on similar topics. What is unique about the perspective that this review provides?
Improvement on the writing is needed. Many sections read like a laundry list of facts, with few transitions, logic, and flow that ties everything together. In some cases, more detail is needed, points are made with no supportive information, such as “the gut microbiota regulates the secretion of mucus by goblet cells”, but not details are provided. Many contemporary references were missed and should be included. .
Check references. Statement about sterile guts at hatch was supported by a 12 yr old review article (ref 18) on probiotics. In the referenced review article, a comment about sterile guts is made, with no citation for a primary article. Please update with a more recent study (there have been some), also please include discussion about microbes on eggshells and succession (vertical transmission).
Line 48 change cecum to ceca, since birds have 2.
The “S“ in “16S” is always capitalized.
Microbiota should be used in place of “microflora”
Line 99, the egg shell may serve as a vector for microbes from the hen to the chick, there is some research on the connection.
In the section about the crop, bacterial fermentation of fiber is discussed. How important is fiber fermentation in the crop compared to that of the distal illeum or ceca?
Perhaps in discussion of each of the gut compartments, the authors could discuss the physiological roles for the host, for context
Ceca section: it is surprising that the authors make a point that Actinobacteria and Proteobacteria make up a small portion of the microbiota, but then list Bifidobacterium and E. coli in the top 3 genera making up the microbiota. Are Bifidobacterium and E. coli among the most abundant genera? A Salmonella paper from 1979 was referenced to support this point, but the authors make the earlier point about how culture studies under represent the diversity of the gut. Please reference modern papers for this point, many currently identified genera that were not named in 1979.
Is there a ban on all AGPs? How about antibiotic not essential for human health, such as bacitracin?
“Stressors are various factors that can induce stress”, shouldn’t define stressor with “stress”
Author Response
Reviewer 2:
We have tried to work on the flow and add the details suggested.
Line 48 change cecum to ceca, since birds have 2- Corrected.
The “S“in “16S” is always capitalized- Line 99- corrected.
Microbiota should be used in place of “microflora”
Line 99, the eggshell may serve as a vector for microbes from the hen to the chick, there is some research on the connection. We have tried to include this in the section from lines 104-121.
“Chicks can acquire microbiota at the embryonic stage during egg formation in the oviduct and during transport through the reproductive tract [21]. Post-hatch microbial acquisition is dependent on various factors such as production practices, diet, and environment. With the modernization of chicken production in large-scale hatcheries, the natural vertical transfer of microbiota from the hen is considerably diminished. Nevertheless, the passage of eggs through the reproductive tract and cloaca will deposit microbiota on the eggshell in the form of a complex biofilm [26]. The obligates anaerobes in the eggshell biofilm can survive the incubation period and can shape the microbial population of the chick. Hatchery sanitation protocols such as washing, fumigation, and chemical disinfection of the egg, though, will reduce the vertical transfer of microbiota. In a study conducted by Olsen et al., [27] it was demonstrated that the bacterial load on eggs decreased from 1×104 - 1×105 CFU to less than 10 CFU per egg. Though disinfection practices are inevitable to ensure optimal hatchability in commercial hatcheries, disinfection impedes the vertical transfer of microbiota. Hence chicks acquire a considerable proportion of gut microbiota from the environment post-hatch.”
In the section about the crop, bacterial fermentation of fiber is discussed. How important is fiber fermentation in the crop compared to that of the distal illeum or ceca? – The physiological role of the different regions of the digestive tract is assumed to be known to the readers of this literature.
“Stressors are various factors that can induce stress”, shouldn’t define stressor with “stress”- rewritten as “Stress can be of dietary, environmental, or managemental origin and can have detrimental effects on the health and production of poultry”.
Reviewer 3 Report
Dear Authors,
This review compiles information on the chicken gut microbiota, dysbiosis, and additives such as probiotics, prebiotics, and synbiotics.
Authors have worked hard and present a very well described paper swhich includes all aspects of microbiota where probiotics play an important role. They postulate a mode of action and effects of probiotic as well as some factors during supplementation and future prospects.
The only thing I could suggest is to add a more extensive discussion on prebiotics and synbiotics, particularly, from the chicken growth, health and performance point of view.
Author Response
Reviewer 3:
The only thing I could suggest is to add a more extensive discussion on prebiotics and synbiotics, particularly, from the chicken growth, health and performance point of view – Revised prebiotics and synbiotics section.
Round 2
Reviewer 1 Report
Thank you for the revisions.
Reviewer 2 Report
Thanks